# Experimental and Numerical Studies on the Direct Shear Behavior of Sand–RCA (Recycled Concrete Aggregates) Mixtures with Different Contents of RCA

**DOI:** 10.3390/ma14112909

**Published:** 2021-05-28

**Authors:** Yiming Liu, Shiqiang Huang, Lihua Li, Henglin Xiao, Zhi Chen, Haijun Mao

**Affiliations:** 1School of Civil Engineering, Architecture and Environment, Hubei University of Technology, Wuhan 430068, China; ymliu@hbut.edu.cn (Y.L.); shiqiang_huang@126.com (S.H.); 20061069@hbut.edu.cn (H.X.); chenzhi1988420@126.com (Z.C.); 2Institute of Rock and Soil Mechanics, Chinese Academy of Sciences, Wuhan 430071, China; hjmao@whrsm.ac.cn

**Keywords:** discrete element method, direct shear test, recycled concrete aggregates (RCA), RCA contents

## Abstract

Recycled concrete aggregate (RCA) is a typical construction and demolition (C&D) material generated in civil engineering activities and has been widely used as the coarse-grained filler added to sand for roadbed fillings. The effect of RCA content on the mechanical behavior of sand–RCA mixtures is complicated and still not fully understood. To explore the effect of RCA content on the macroscale and microscopic behavior of the sand–RCA mixtures with various RCA contents, laboratory direct shear tests and numerical simulations using the 3D discrete element method were performed. Experimental direct shear tests on sand–RCA mixtures with different contents of RCA were first carried out. Numerical direct shear models were then established to represent the experimental results. The particle shape effect was also considered using a new realistic shape modeling method to model the RCA particles. Good agreement was observed between the DEM simulation and experimental results, verifying the ability of the numerical direct shear models to represent the direct shear behavior of sand–RCA mixtures. The macroscopic responses of both experimental and numerical tests showed that all samples presented an initial hardening followed by a post-peak strain softening. The peak-state friction angles increased with the RCA content for samples under the same vertical stress. The effect of RCA content on the microscopic behavior based on DEM simulation was also found. The microscopic properties of RCA–sand mixtures, such as coordination numbers, PDFs and contact force transformation features, were analyzed and related to the macroscopic results.

## 1. Introduction

With the rapid urbanization and the ever-increasing population worldwide in recent years, an enormous amount of construction and demolition waste has been produced due to the ongoing civil engineering activities [1,2,3,4], which is causing serious environmental problems and threatening the sustainability of the modern society [5,6,7]. Therefore, the reuse of construction and demolition waste has attracted increasing attention globally. Recycled concrete aggregate (RCA) is one of the most favorable and commonly implemented reutilization forms of construction and demolition waste [8]. Recycled concrete aggregate (RCA) is obtained by crushing the demolition waste of aged concrete buildings [7,9]. Previous research shows that the engineering properties of RCA are quite different from those of natural aggregates [10]. Compared to natural aggregates, RCA usually has a more angular shape and rougher surface [11,12,13]. These features are believed to be able to increase the friction angle of RCA [10]. RCA has been sustainably applied as substitute for natural aggregates in transportation and geotechnical projects [2,6,12,14,15,16,17,18,19].

RCA can also be used as coarse aggregates added to geomaterials with fine particles, which would improve the mechanical properties of the fine geomaterials and thus save the limited natural aggregate sources. Many researchers have investigated the performance of RCA as coarse aggregates blended with different types of fine particles. For example, Yaghoubi et al. [20] studied the suitability of RCA blended with polyethylene granules and found that the strength properties of the mixture were comparable with those of natural aggregates. Aruljarah et al. [14] investigated the mechanical behavior of RCA mixed with different contents of fine recycled municipal glass (FRG) and showed that the mixture consisting of 85% RCA and 15% FRG had a promising result for strength. Li et al. [21] studied the shear strength of different proportions of sand mixed with RCA particles and found that the addition of RCA particles can significantly improve the shear strength of the sample. RCA mixed with fine particles forms a typical binary mixture. According to previous research, the properties of binary mixtures with coarse aggregates and fine particles are mainly determined by the content and properties of the coarse aggregates [22,23]. Many researchers have paid attention to the effect of coarse aggregate on the mechanical performance of the binary mixture using triaxial tests [24,25,26], direct shear tests [27,28] and simple shear tests [29,30]. Another tool to study the mechanical behavior of binary mixtures is the discrete element method. Comparing to laboratory tests, the discrete element method is able to not only capture the macroscopic behavior of binary mixtures but also study the particle-scale characteristics of binary mixtures [31]. Many effects have been investigated to analyze the macro- and micro-behavior of binary mixtures with the discrete element method [30,32,33,34,35]. These numerical results can represent the laboratory results and have significantly improved the comprehension of the mechanical behavior of binary mixtures.

Although the aforementioned studies have reported some findings, comprehensive studies on the effect of RCA content on mechanical behavior of sand–RCA mixtures are still rare. Especially, an investigation from a microscopic view considering the particle shape has not been reported. This study conducted a systematic investigation of the effect of RCA content on the shear behavior of sand–RCA mixtures using laboratory tests and 3D DEM simulation. To do so, large-scale direct shear tests were first carried out on samples with various RCA contents ranging from 0% to 80% under different confining stresses. Then, numerical models were established to simulate experimental tests. RCA particles with authentic shapes were generated using a new method based on the 3D reconstruction technique. The numerical results were calibrated with the experimental results to determine the microscopic parameters. After that, the effect of RCA content on the microscale properties of RCA–sand mixtures, such as the contact network, coordination number and contact force distribution was elucidated and analyzed. Based on the microscopic behavior, the reasons for the effect of RCA content on the shear strength of the sand–RCA mixtures were also analyzed. The findings of this study enhance the knowledge of the mechanical performance of sand–RCA mixtures, which is likely to benefit the engineering practice.

## 2. Laboratory Tests

### 2.1. Materials

Sand–RCA mixtures consist of small sand particles and large RCA particles. In this study, the RCA particles were produced by crushing blocks of demolished structural concrete obtained in an abandoned building in Wuhan, China. The maximum particle size of RCA was 37.5 mm, while that of sand particles was 5 mm. The particle size distributions of sand and RCA particles are presented in Figure 1. The densities of sand and RCA particles were 2.70 and 2.50 g/cm^3^, respectively.

### 2.2. Test Plan

Direct shear tests were conducted using a large-scale direct shear apparatus with a size of 300 mm × 300 mm × 300 mm, as shown in Figure 2. The ratio of the direct shear dimension to the maximum particle size was greater than 7, which satisfies the sample size to particle size criterion. Sand and RCA particles were first mixed uniformly and then poured into the direct shear equipment with three layers. Each layer was compacted to a given porous ratio. Finally, the shear process was conducted by moving the lower box with a shear rate of 0.01 mm/s until the shear displacement reached 30 mm, or 10% of the shear strain. In this study, sand–RCA mixture samples were prepared with the five RCA contents of 0%, 20%, 40%, 60% and 80%. The direct shear tests were carried out with the three vertical stresses of 100, 200 and 300 kPa. The densities of samples with different RCA contents are listed in Table 1.

## 3. DEM Simulation of Direct Shear Test

### 3.1. Modeling Authentic RCA

Particle shape plays an important role in the mechanical performance of granular materials. Many studies using DEM have been performed to investigate the effect of particle shape. To establish a realistic particle model by using DEM, finding the particle morphology is the first step. X-ray scanning and laser scanning are two of the most used methods to establish the morphology of realistic particles [36,37,38,39,40,41]. However, special instruments are necessary when applying these two methods. In this study, we proposed a new method to model realistic particles without special equipment. The new method used in this study consists of three steps, as shown in Figure 3. In the first step, typical particles are selected and photo are taken from different directions. In the second step, the photos are converted into a 3D model using 3ds Max as a STL-file. In the third step, the STL-files are imported into PFC3D 5.0, and clumps with authentic shapes are generated by means of the bubble-pack method in PFC3D 5.0 based on the imported STL-files. Figure 4 presents nine typical 3D model of RCA and the associated DEM clumps.

### 3.2. Numerical Direct Shear Model

The commercial discrete element method software PFC3D was applied to conduct the numerical direct shear tests in this study. The numerical direct shear model consisted of 10 rigid walls with the same size as the laboratory instrument. In DEM simulations, one of the most important factors that influences the computation efficiency is the particle number. The particle number of binary mixtures with coarse and fine particles is usually very big, which will significantly increase the computational time. To overcome this limitation, the upscaling method is widely used by expanding the particle size. This scaling technology was also applied in this study. The particles size distributions of the numerical samples are shown in Figure 5.

Previous numerical studies usually simulated coarse particles with realistic geometries and fine particles as spheres when conducting experiments on binary mixtures with coarse particles. In this study, the same strategy was applied to model the sand–RCA mixtures. The RCA particles were generated with realistic RCA geometries by the built-in clump logic using the nine clump templates presented in Section 3.1. The sand particles were generated as spherical finer particles because the shape of sand particles was more regular and the size of sand particles was much smaller than those of the RCA particles. A very low friction coefficient was assigned to the particles to decrease the particle interlocks and make the system quickly move to an equilibrium state. After that, the numerical model was consolidated to given confining pressures by moving the top wall controlled by a servo mechanism. The confining stresses used in this study were 100, 200 and 300 kPa. Then, the numerical model was sheared by moving the lower box with a steady rate of 0.1 mm/s until the shear displacement reached 30 mm. The numerical direct shear test model with 40% RCA is shown in Figure 6.

### 3.3. Contact Model and Micro-Parameters

The linear contact model was applied to represent the RCA–RCA contact behavior, while the built-in rolling resistance contact model was applied for sand–sand contacts. The reason for this choice is that the RCA particles were generated with realistic shapes, while the sand particles were spheres and did not consider the shape effect. The rolling resistance model was reported to be a good alternative method considering the particle shape effect.

The linear contact model has two components between the two contacting entities: the normal component and the shear component. The constitutive behavior of the normal component can be described as
(1)$$F^{n}=K^{n}U^{n}$$
where $F^{n}$ is the normal contact force, $K^{n}$ is the normal stiffness and $U^{n}$ is the overlap.

The constitutive behavior of the shear component is described in incremental mode:(2)$$\Delta F^{s}=-K^{s}\Delta U^{s}$$
where $\Delta F^{s}$ is the normal contact force, $K^{s}$ is the normal stiffness and $\Delta U^{s}$ is the incremental tangential displacement.

The contact behavior also follows a slip model. Contact slip is allowed to occur, when the maximum shear contact force $F_{max}^{s}$ satisfies the slip condition as follows:(3)$$F_{max}^{s}\geq \mu F^{n}$$
where μ is the friction coefficient.

The rolling resistance model is developed based on the linear contact model by adding a moment $M^{r}$ to resist the relative rotation between the contact entities. The rolling resistance moment can be expressed as follows:(4)$$\Delta M^{r}=-k_{r}\Delta \theta_{b}$$
where $M^{r}$ is the rolling resistance moment, $k_{r}$ is the rolling resistance stiffness and $\theta_{b}$ is the relative rotation increment.

In PFC3D, $k_{r}$ is defined as
(5)$$k_{r}=k_{r}{\bar{R}}^{2}$$
where $\bar{R}=R^{\left(1\right)}\;R^{\left(2\right)}/\left(R^{\left(1\right)}+R^{\left(2\right)}\right)$ is the contact effective radius and $R^{\left(1\right)}\;\mathrm{and}\;R^{\left(2\right)}$ are the radii of the contact entities.

The magnitude of the rolling resistance moment should satisfy a rolling slip model.
(6)$$M^{r}=\left\{\begin{array}{c} M^{r}\;,\left|M^{r}\right|<M^{*}\;\\ M^{*}\left(\frac{M^{r}}{\left|M^{r}\right|}\right),\;otherwise\; \end{array}\right.$$

The limiting torque $M^{*}$ can be calculated as follows:(7)$$M^{*}=\mu^{r}\bar{R}F^{n}$$
where $\mu^{r}$ is the rolling friction coefficient.

To obtain the micro-parameters of both RCA and sand particles, a calibration process was conducted in this study. The numerical pure sand samples were first calibrated with the experimental results to get the micro-parameters of sand particles. Then, the micro-parameters of RCA particles were obtained by matching the numerical performance of numerical sand–RCA mixtures with the laboratory tests results. The micro-parameters obtained in this study are shown in Table 2. The stress–strain behaviors of both numerical results and experimental data were compared and presented in Figure 7. It is observed that the numerical results were consistent with the experimental data.

## 4. Results Analyses

### 4.1. Macroscopic Behavior

Figure 7 presents the evolution of shear stress versus shear strain for samples with different RCA contents under confining pressures of 100, 200 and 300 kPa. The numerical simulation results and experimental data are both demonstrated and compared. It is observed that the macroscopic behavior of all specimens with various RCA content shows an initial strain hardening and then post-peak softening behavior. The shear stress for each sample initially increases with shear–strain until reaching the peak state, and then the shear stress slightly decreases with the shear strain. The shear strain in the peak state also increases with the confining stress for all specimen with various RCA contents. It is also observed that the shear stress in the peak state also increases with the confining stress.

To illustrate the effect of RCA content on the stress–strain behavior of sand–RCA mixtures, the curves of shear stress versus shear strain for samples with various RCA contents under the confining stress of 200 kPa are plotted and demonstrated in Figure 8. Similar to Figure 7, both the experimental results and numerical data are demonstrated in Figure 8. As shown in Figure 8, the curves of shear stress for the five samples show a similar trend. The shear stresses of the five samples follow the stress softening pattern. It is also observed that the evolution of shear stress is affected by the RCA content. A sample with a higher RCA content has a higher shear stress at the same shear strain.

The peak shear stresses for samples with various RCA content under different confining stresses are shown in Figure 9. For comparison, the peak shear stresses obtained from experimental and numerical results are both presented. As shown in Figure 9, the peak shear stress is affected by the RCA content. The peak shear stress increases with RCA content under different confining stresses. This can be explained as follows: with an increase of RCA content, the interlocking between particles increases due to the coarse size and irregular shape of the RCA particles. It should be noted that, with the increase of confining stress, the deviations between the numerical results and experimental data increase, as shown in Figure 7 and Figure 9. This phenomenon may be attributed to particle breakage, which was not considered in this study. According to previous research, particle breakage can significantly affect the mechanical behavior of granular materials, and the degree of particle breakage increases with confining pressures.

### 4.2. Microscopic Behavior

In this section, the effect of RCA content on the microscopic behavior of sand–RCA mixtures, i.e., the coordination number, contact forces and their distribution, are discussed.

#### 4.2.1. Coordination Number

The coordination number is an important microscopic parameter closely relevant to the internal stability of contact networks [42,43]. The overall coordination number ${\mathrm{CN}}_{\mathrm{ALL}}$ is usually defined as follows:(8)$$CN_{ALL}=2N_{c}/N_{p}$$
where $N_{c}$ is the number of total contacts and $N_{p}$ is the number of particles.

Figure 10 shows the evolutions of the coordination numbers of samples with various RCA contents under the confining stress of 200 kPa. Figure 10 illustrates that the RCA content plays an important role in the evolution of the overall coordination number $CN_{ALL}$. With the increase of RCA content, the RCA particles dominate the sand–RCA mixtures and the overall coordination number $CN_{ALL}$ also increases.

Sand–RCA mixtures are typical binary mixture materials with three types of contacts: RCA–RCA, sand–sand and RCA–sand. In the present study, the coordination numbers for these three types of contacts were also analyzed. The definitions of these three coordination numbers followed those of Minh and Cheng [44] and can be calculated as
(9)$$CN_{ss}=2N_{c,s-s}/N_{p,s}$$
(10)$$CN_{RS}=2N_{c,R-s}/N_{p}$$
(11)$$CN_{RR}=2N_{c,R-R}/N_{p,R}$$
where $CN_{ss}$ is the sand–sand coordination number, $CN_{RS}$ is the RCA–sand coordination number, $CN_{RR}$ is the RCA–RCA coordination number, $N_{c,s-s}$ is the number of sand–sand contacts, $N_{c,R-s}$ is the number of RCA–sand contacts, $N_{c,R-R}$ is the number of RCA–RCA contacts and $N_{p,s}$ and $N_{p,R}$ are the numbers of sand and RCA particles, respectively.

The evolutions of $CN_{ss}$, $CN_{RR}$ and $CN_{RS}$ versus shear strain are shown in Figure 11. In Figure 11a,b, the coordination number of RCA–RCA contacts increases with the RCA content, while the coordination number of sand–sand contacts decreases. The coordination number of RCA–sand contacts increases with the RCA content until the RCA content reaches 60%, and then it decreases with the RCA content.

#### 4.2.2. Contact Forces in the Peak State

To thoroughly explore the effect of RCA content on the force transmission, the visualization of force chain distribution is necessary. Figure 12 shows the contact force distributions of samples with various RCA contents in the peak state under confining stress of 200 kPa. Each contact force is exhibited as a color line segment with a thickness proportional to its magnitude in Figure 12. Comparing these five specimens, the density of contact force distribution decreases with the increase of RCA content, especially for the contact forces with low magnitudes. This may be explained as follows: with the increase of RCA content, the particle number decreases and forces transport through fewer contacts.

Figure 13 presents the relationships between RCA contents and average normal contact forces of samples under different confining stresses in the peak state and the final state. For samples under the same confining stress, the average normal contact forces increase with RCA contents in both the peak state and the final state, while, for samples with the same RCA content, the average normal contact forces increase with confining stresses. This is because that the particle size of RCA is much larger than that of sand. With the increase of the RCA content, the total number of the sample decreases, the shear strength increases and the contact force taken by each contact increases.

The contact force distributions of samples with different RCA contents, as shown in Figure 12, gave a qualitative presentation of the characteristics of the contact force network in these sand–RCA mixtures. To quantitatively investigate the effect of RCA content on the characteristic of the contact forces, an appropriate mathematical method is necessary. The probability density function (PDF) method, which is widely used to characterize contact force transmission, was applied in this study. The contact forces can be divided into strong contact forces and weak contact forces based on whether one contact force is larger or smaller than the average contact force. A contact force which is larger than the average contact force is called a strong contact force, otherwise it is called a weak contact force. The probability of strong and weak contact forces can be expressed by two different equations as follows:(12)$$P\left(f_{x}\right)=\left\{\begin{array}{c} e^{-A\left(\frac{\left|f_{x}\right|}{\langle \left|f_{x}\right|\rangle }\right)},\;f_{x}>\langle \left|f_{x}\right|\rangle \;\\ {\left(\frac{\left|f_{x}\right|}{\langle \left|f_{x}\right|\rangle }\right)}^{B},\;f_{x}\leq \;\langle \left|f_{x}\right|\rangle \; \end{array}\right.$$
where $f_{x}$ is the normal contact force or shear contact force, $\langle \left|f_{x}\right|\rangle$ is the average value of $\left|f_{x}\right|$ and A and B are coefficients related to the inhomogeneity of contact forces.

Figure 14 shows the probability density function (PDF) of the contact forces in the peak state for samples with different RCA contents under confining stress of 200 kPa. As shown in Figure 14, the probability densities of the two types of contact forces are both significantly affected by the RCA content. It can be observed that, for a given value of contact force, the probability density increases with RCA content, while, for a given value of probability, the contact force increases with RCA content. Figure 15 demonstrates the PDFs of normal contact forces and shear contact forces normalized by the average normal contact force and the average shear contact force for samples with different RCA contents under confining stress of 200 kPa. As shown in Figure 15, the PDFs are nearly the same for both normalized normal and shear contact forces when the normalized contact force is less than a certain value. This value for the normalized normal contact force is 5.0, while that of the normalized shear contact force is 7.0. This may be attributed to the angular shape and size of the RCA particles. With the increase of the RCA content, the RCA particles dominate the force transformation of the samples. Since the RCA particles have a larger size and a more angular shape, the interlocking between RCA particles will increase with the increase of the RCA content. This phenomenon also explains why the shear strength increases with the increase of the RCA content.

#### 4.2.3. Contact Force Anisotropy

The effect of RCA content on contact force transmission during different shear states was also explored in this study. Figure 16 and Figure 17 illustrate the anisotropy of normal and shear contact forces of samples in the initial state, peak state and the final state in polar coordinates. As shown in Figure 16, the normal contact force chains first rotate clockwise for all samples with different RCA contents until the shear strain reaches the peak state, and then they slightly rotate anticlockwise in the final state. This observation means that the anisotropy of normal contact force increases from the initial state to the peak state and then decreases slightly after the peak state. The shear contact force transmission shown in Figure 17 illustrates that the anisotropy of shear contact force seems to not be affected during the shear process, although the magnitudes of the shear contact forces are quite different.

To explore the effect of RCA contents on the contact force transmission features, quantitative studies were conducted to investigate the magnitude and direction of the anisotropy for contact force chains in RCA–sand mixtures with different RCA contents. The shear-induced anisotropy of contact forces is an important feature to illustrate the microscopic force distribution of granular materials. In DEM simulations, the second-order Fourier series approximation (FSA) proposed by Rothenburg et al. [45] is usually used to describe the anisotropy features.

The main idea of the FSA method is that the directional distribution of micro-quantities can be described by using a probability density function, $E\left(\theta \right)$, such as the second-order form of Fourier series expansion.
(13)$$E\left(\theta \right)=E_{0}\left[1+\Delta d\cos2\left(\theta -\theta_{d}\right)\right]$$
where $E_{0}$ represents an isotropic distribution part and Δd and $\theta_{d}$ are the density and principal direction of the anisotropic distribution of a micro-quantity, respectively.

The anisotropic distribution of the normal contact force can also be described by using an equation similar to Equation (13). The equation for normal contact force distributions is given as follows:(14)$$F_{n}\left(\theta \right)=F_{n}\left\{1+\alpha_{n}\cos\left[2\left(\theta -\theta_{n}\right)\right]\right\}$$
where $F_{n}\left(\theta \right)$ is the sum of normal contact forces in interval θ, $F_{n}$ is the average of the normal contact force in all intervals,$\;\alpha_{n}$ is the normal contact force anisotropy coefficient and $\theta_{n}$ is the principal direction angle of the normal force anisotropy.
(15)$$F_{n}=\int_{0}^{2\pi }F_{n}\left(\theta \right)d\theta$$

The distribution of the shear contact force is different from that of the normal contact force and has four symmetrical peaks, as shown in Figure 17. The expression to approximate the shear force distribution is given as follows:(16)$$F_{t}\left(\theta \right)=-F_{n}\alpha_{t}\sin\left[2\left(\theta -\theta_{t}\right)\right]$$
where $\alpha_{t}$ is the tangential contact force anisotropy coefficient and $\theta_{t}$ is the principal direction angle of the shear force anisotropy.

Table 3 and Table 4 present the quantitative analysis results of the anisotropy of the normal and shear contact forces of samples with various RCA contents under the shear strains of 0%, 5% and 10%. As shown in Table 3, the values of $\theta_{n}$ for samples with various RCA contents are all about 90° at the shear strain of 0%, which is in line with the theoretical direction of the anisotropy. The values of $\theta_{n}$ for samples with various RCA contents range from 45° to 52° at the shear strain of 5% and from 55° to 61° at the shear strain of 10%. This means that the normal contact force anisotropy of samples with different RCA contents is related to the shear strain. It was also found that $\theta_{n}$ decreases with the increase of RCA content at the shear strains of 5% and 10%, which means that the normal contact force anisotropy reduces when the RCA content increases. The magnitude of normal contact force anisotropy seems to not be affected by the RCA content. The results in Table 4 show that the effect of RCA content on the shear contact force anisotropy is different from that on the normal contact force anisotropy. It is shown that $\theta_{t}$ seems to be independent of the RCA content, while the magnitude of shear contact force anisotropy increases with the RCA content.

## 5. Conclusions

In this study, the mechanical behavior of RCA–sand mixtures was investigated by using the direct shear test and the 3D discrete element method. The experimental direct shear tests on RCA–sand mixtures with different RCA contents were first carried out. Then, numerical direct shear models of RCA–sand mixtures were established using the 3D discrete element method and calibrated with the experimental results. The DEM simulation results are in good agreement with the experimental results, suggesting the validity of the DEM model to investigate the mechanical performance of RCA–sand mixtures in direct shear tests. The microscopic properties of RCA–sand mixtures, such as coordination numbers, PDFs and contact force transformation features, were also analyzed and related to the macroscopic results. The main conclusions drawn from this study are listed as follows:The RCA content plays an important role in the mechanical performance of RCA–sand mixtures. Both the laboratory test and numerical simulation results show that the shear strength increases after adding RCA to sand, and, with the increase of the RCA content, the shear strength of the RCA–sand mixtures increases.The coordination number of all contacts $CN_{ALL}$ increases with the RCA content. For specific contacts, the $CN_{SS}$ decreases and the $CN_{RR}$ increases with RCA content, while the $CN_{RS}$ has an increasing trend with the RCA content increasing from 20% to 60%, and then it decreases as the RCA content increases to 80%.The distribution of contact force can be significantly affected by the RCA content. An increase in the RCA content leads to an increase in the average contact force. Analyses of the PDFs of contact forces show that RCA tends to have a more significant effect on the occurrence of large contact forces.The RCA content can significantly affect the anisotropies of normal contact forces. The rotation of the normal contact force anisotropy $\theta_{n}$ decreases with the RCA content. Meanwhile, the anisotropies of the shear contact forces seem to be much less affected by the RCA content, while the rotation of the shear contact force anisotropy $\theta_{t}$ was found to be independent of the RCA content.

## Figures and Tables

**Figure 1 materials-14-02909-f001:**
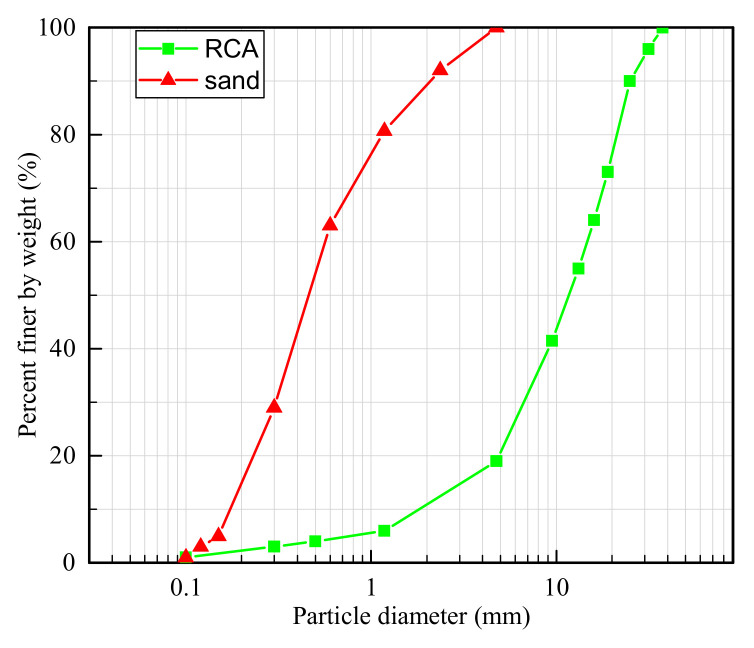
Particle size distributions of sand and RCA.

**Figure 2 materials-14-02909-f002:**
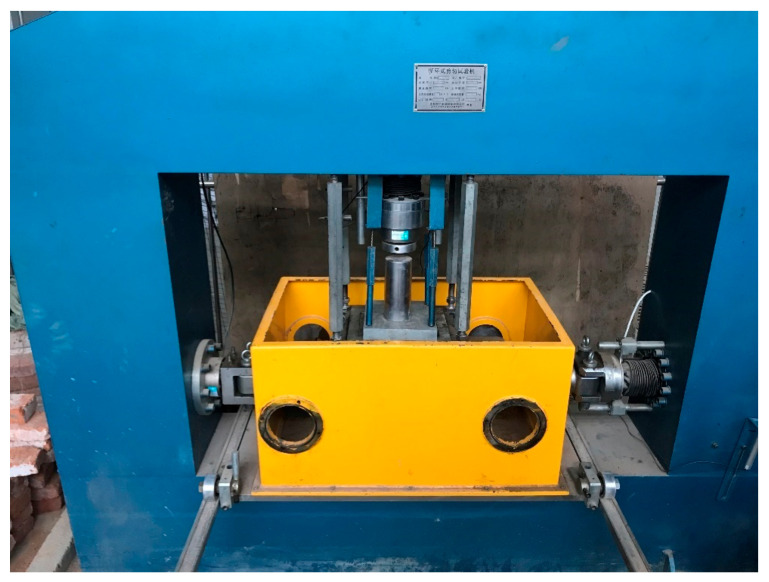
Large-scale direct shear apparatus.

**Figure 3 materials-14-02909-f003:**
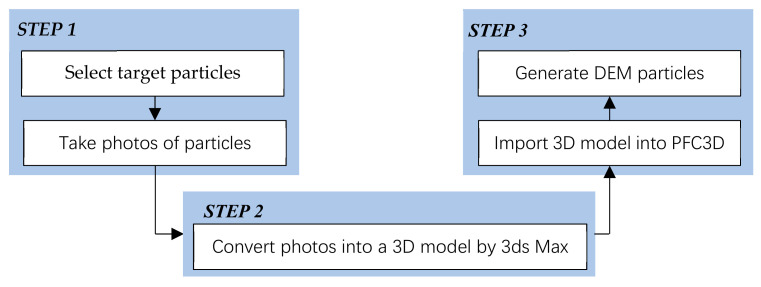
The flowchart of the new method of generating DEM particles with authentic shapes.

**Figure 4 materials-14-02909-f004:**
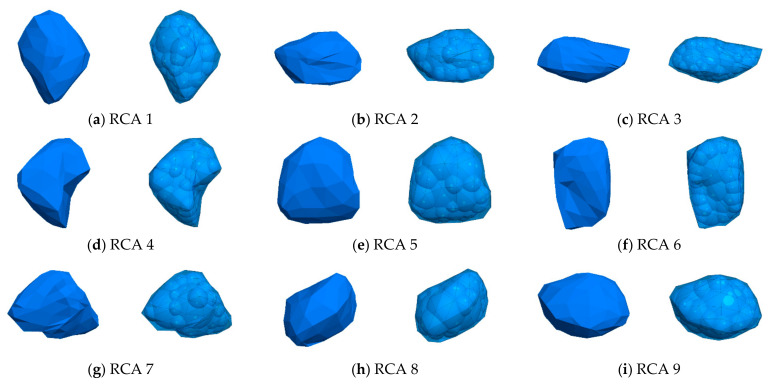
Nine typical RCA aggregates.

**Figure 5 materials-14-02909-f005:**
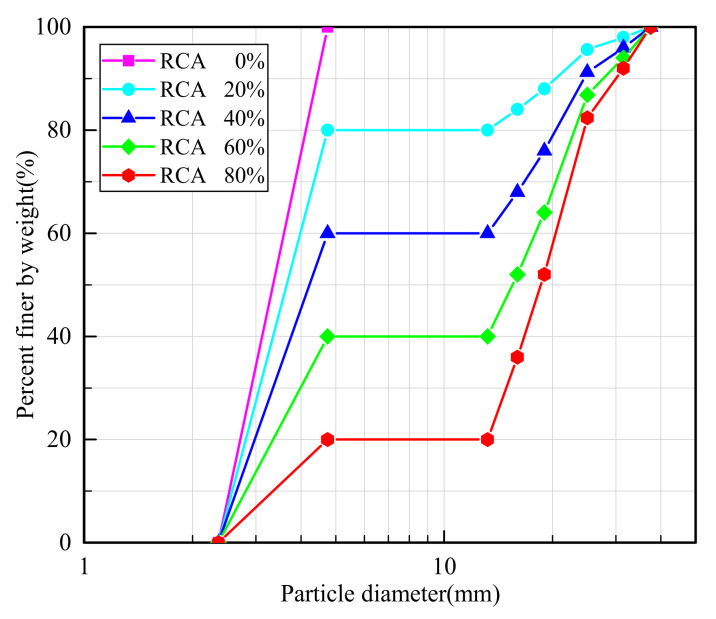
Particle size distributions of the numerical samples with different RCA contents.

**Figure 6 materials-14-02909-f006:**
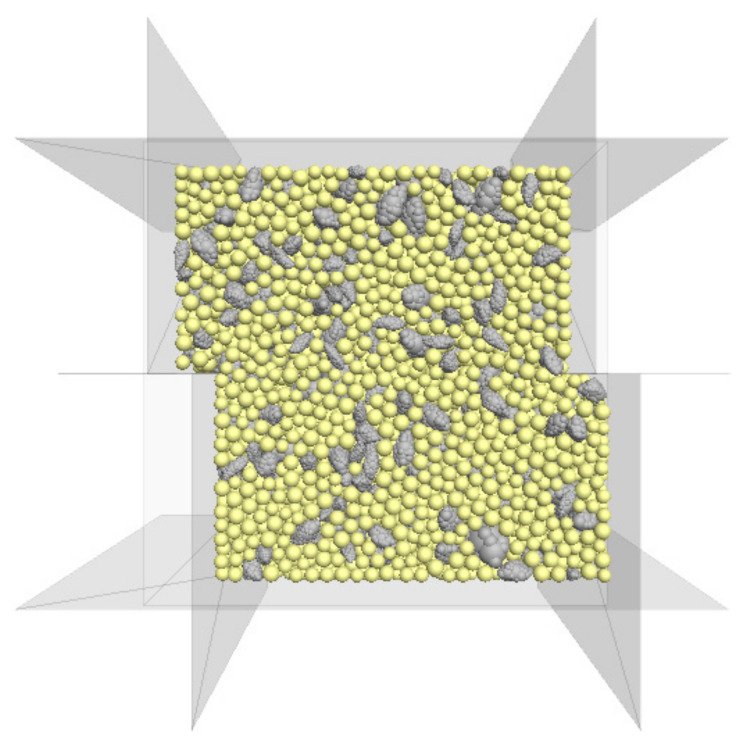
The numerical direct shear model of a sand–RCA mixture with 40% RCA particles.

**Figure 7 materials-14-02909-f007:**
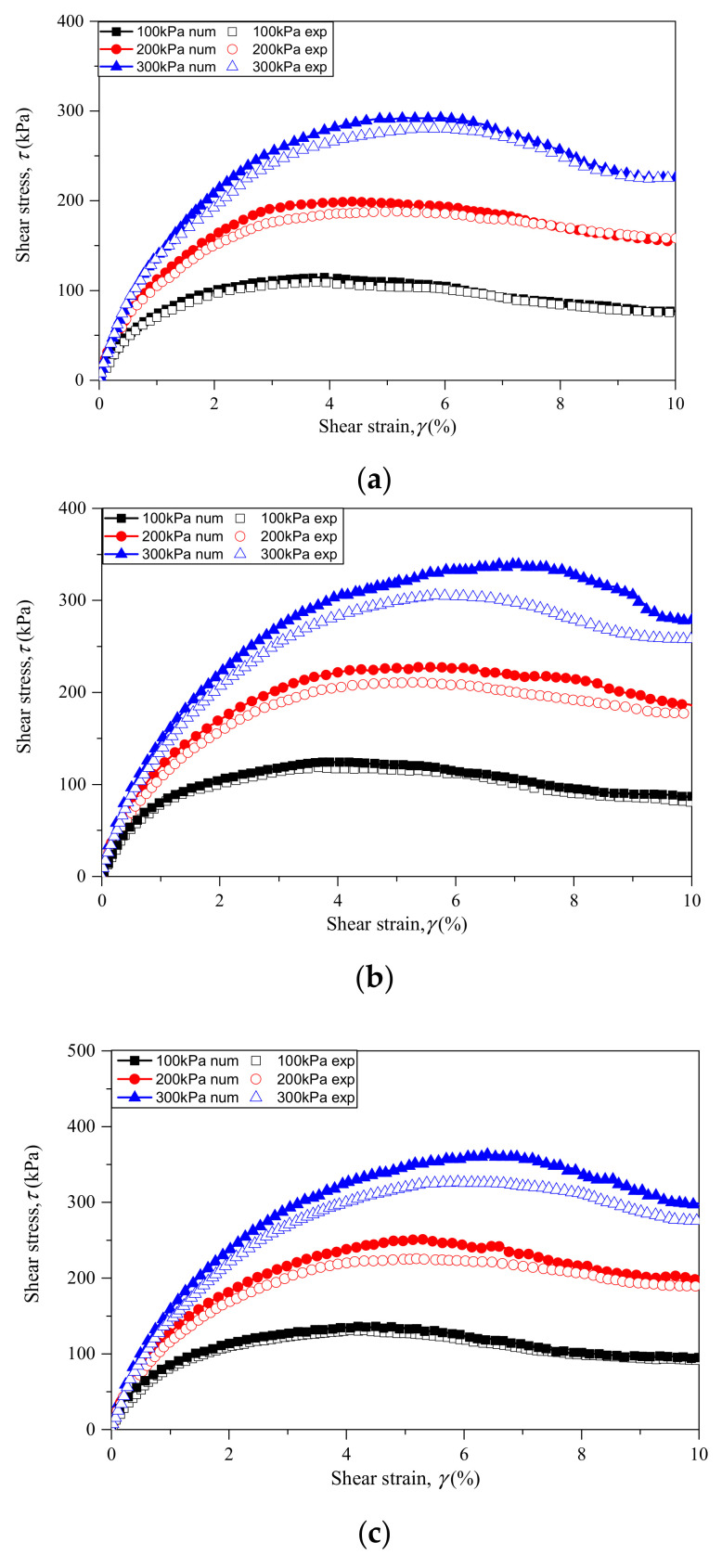
Stress–strain behavior of sand–RCA mixtures under different confining stresses with RCA contents of: (**a**) 0%; (**b**) 20%; (**c**) 40%; (**d**) 60%; and (**e**) 80%.

**Figure 8 materials-14-02909-f008:**
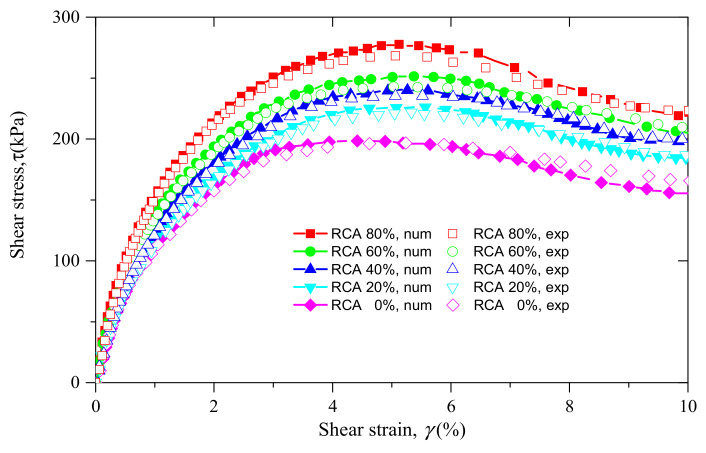
The curves of shear stress versus shear strain of the experimental samples and the numerical models with various RCA contents under the confining stress of 200 kPa.

**Figure 9 materials-14-02909-f009:**
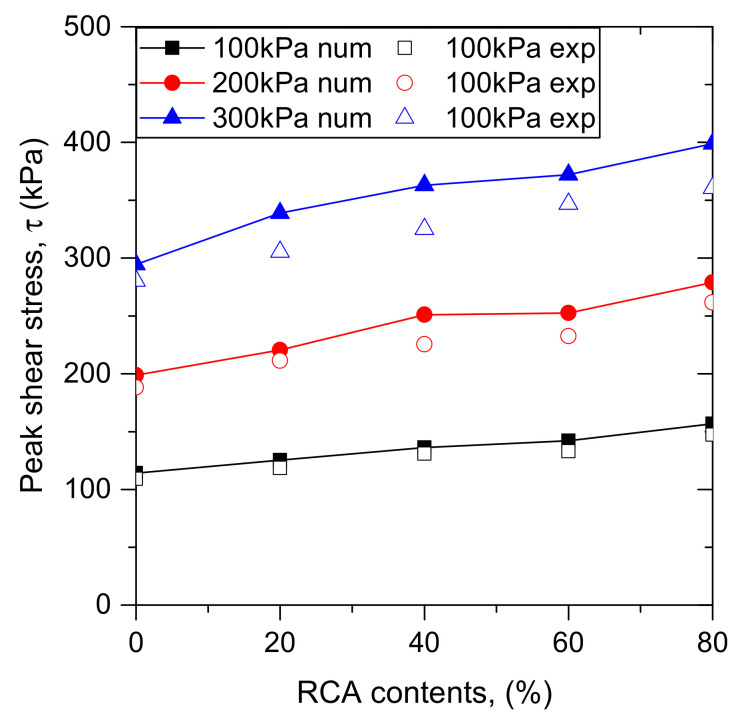
Relationship between peak shear stress and RCA content.

**Figure 10 materials-14-02909-f010:**
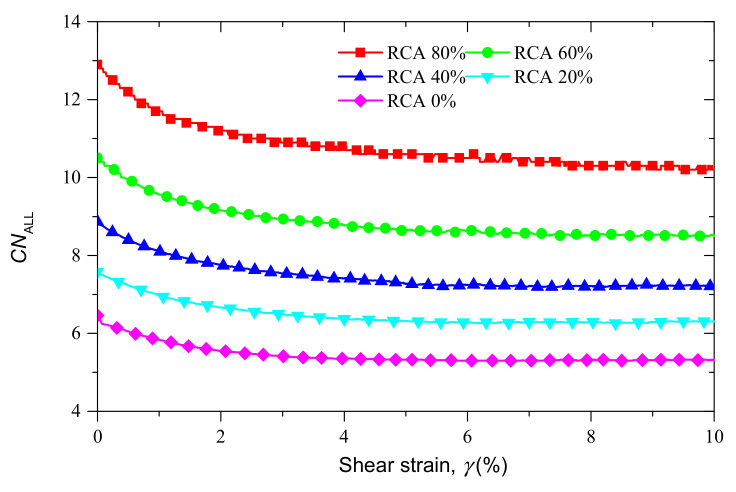
Coordination number versus shear strain for samples with different RCA contents under confining stress of 200 kPa.

**Figure 11 materials-14-02909-f011:**
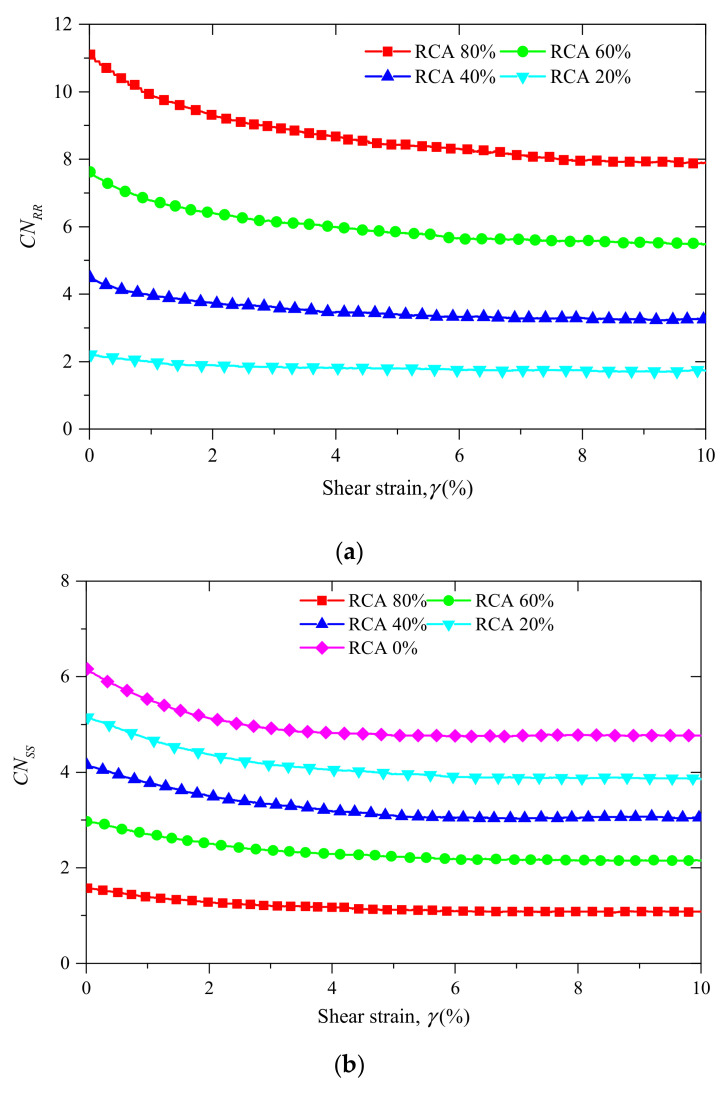
The evolutions of coordination numbers for samples with different RCA contents under confining stress of 200 kPa: (**a**) RCA–RCA contacts; (**b**) RCA–sand contacts; and (**c**) sand–sand contacts.

**Figure 12 materials-14-02909-f012:**
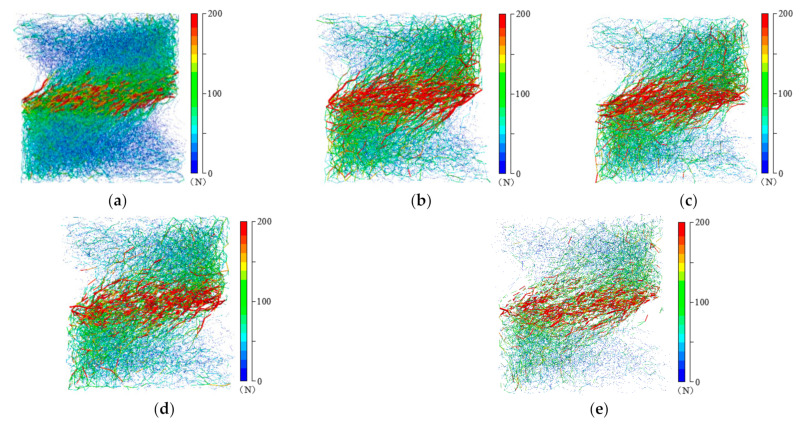
Contact networks of samples with different RCA contents in the peak state under confining stress of 200 kPa: (**a**) RCA content = 0%; (**b**) RCA content = 20%; (**c**) RCA content = 20%; (**d**) RCA content = 20%; (**e**) RCA content = 20%.

**Figure 13 materials-14-02909-f013:**
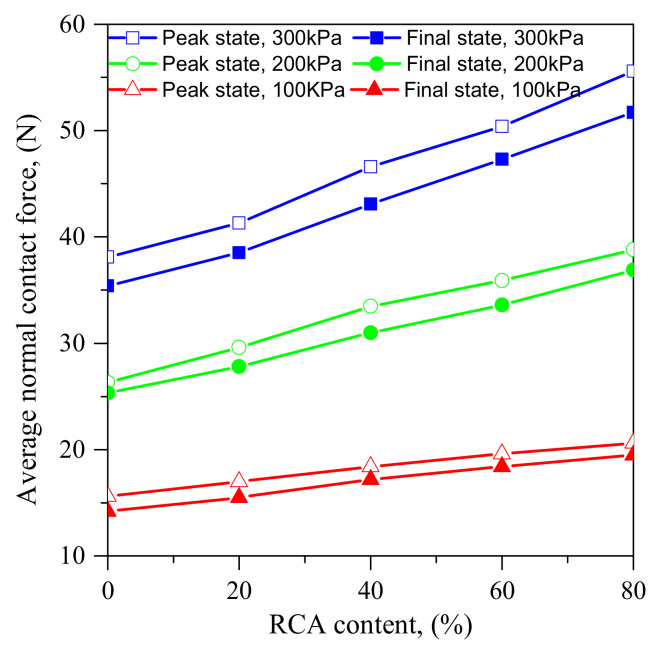
Relationships between RCA content and average contact forces of samples under various confining stresses in the peak state and final state.

**Figure 14 materials-14-02909-f014:**
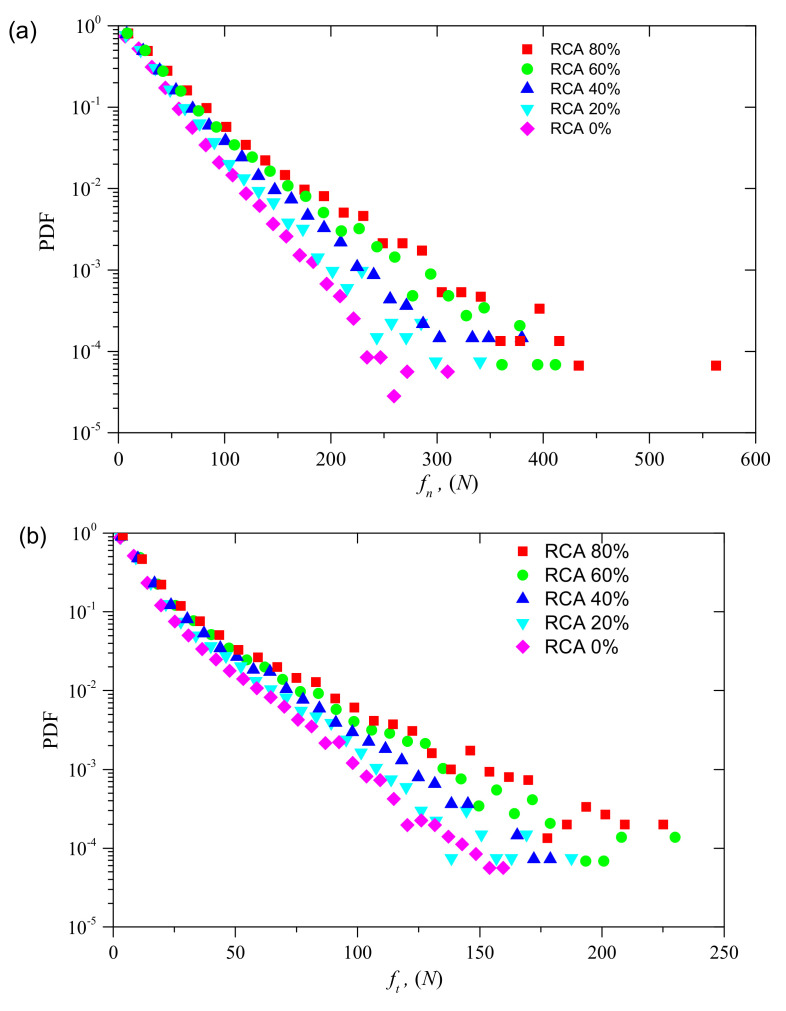
PDFs of samples with different RCA contents: (**a**) normal contact force; and (**b**) shear contact force.

**Figure 15 materials-14-02909-f015:**
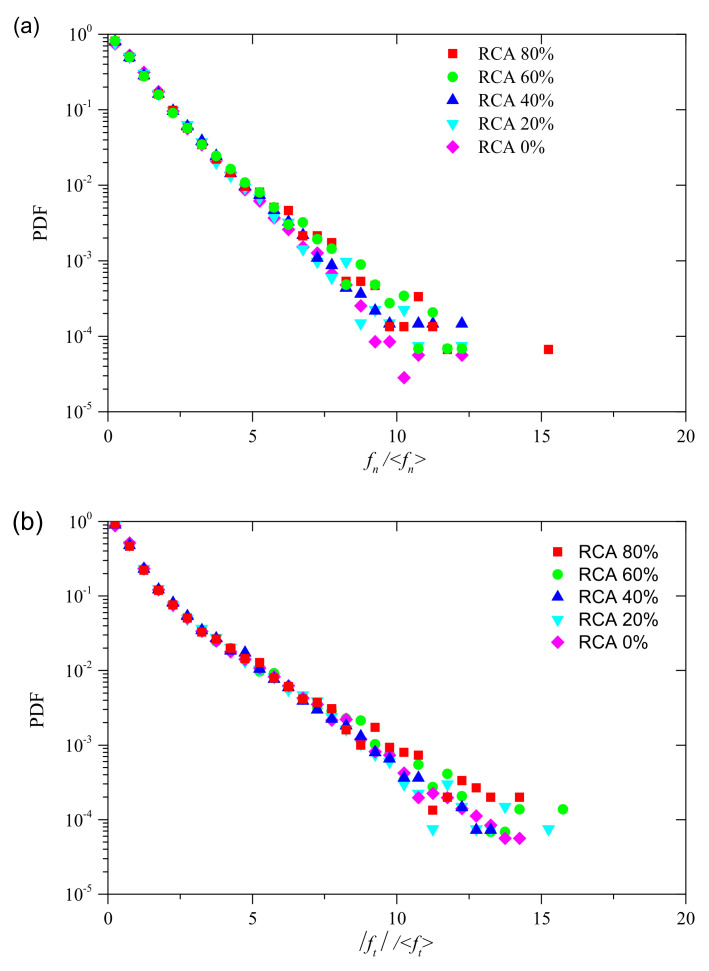
PDFs of samples with different RCA contents: (**a**) normalized normal contact force; and (**b**) normalized shear contact force.

**Figure 16 materials-14-02909-f016:**
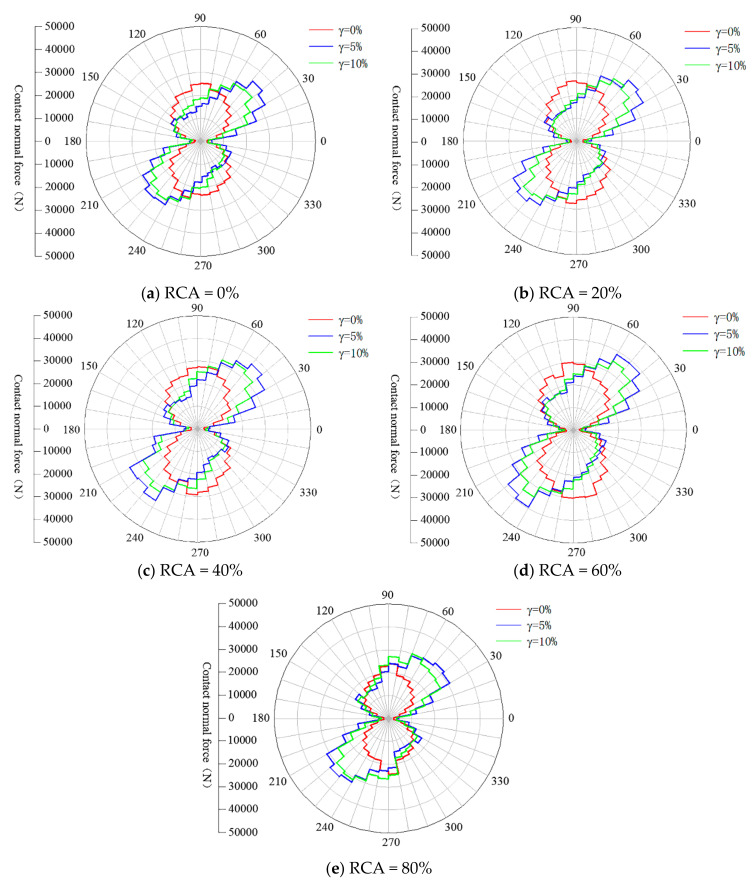
The anisotropies of normal contact force of samples with various RCA contents in different shear states: (**a**) RCA = 0%; (**b**) RCA = 20%; (**c**) RCA = 40%; (**d**) RCA = 60%; and (**e**) RCA = 80%.

**Figure 17 materials-14-02909-f017:**
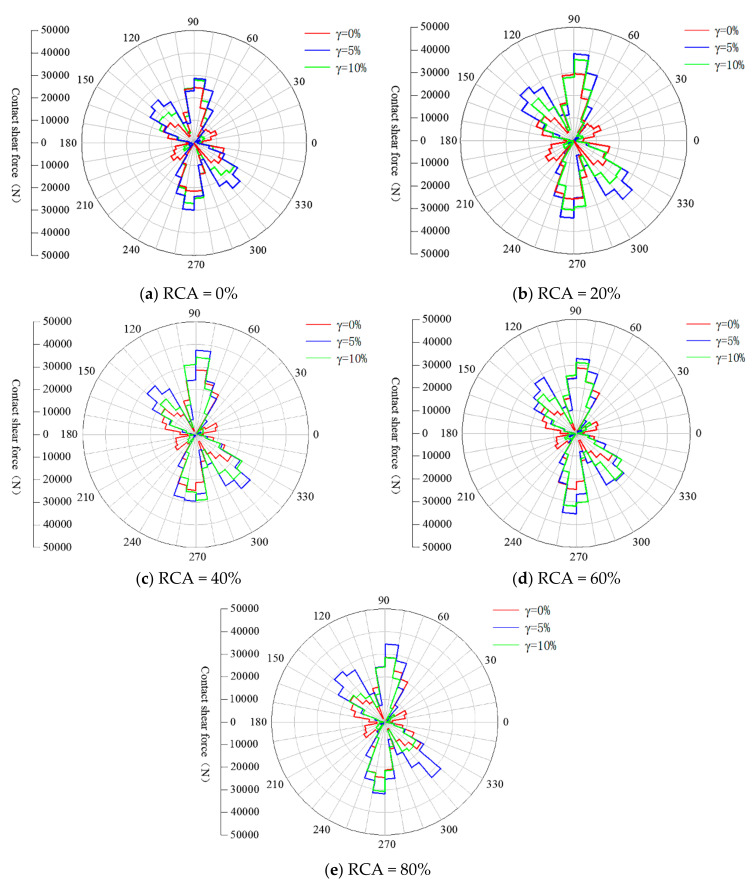
The anisotropies of shear contact force of samples with various RCA contents in different shear states: (**a**) RCA = 0%; (**b**) RCA = 20%; (**c**) RCA = 40%; (**d**) RCA = 60%; and (**e**) RCA = 80%.

**Table 1 materials-14-02909-t001:** Densities of samples with different RCA contents.

RCA Content	0%	20%	40%	60%	80%
Density (kg/m^3^)	1720	1742	1764	1786	1808

**Table 2 materials-14-02909-t002:** Microscopic parameters used in this study.

Parameters	RCA	Sand	Wall
Density, *ρ* (kg/m^3^)	2500	2700	-
Normal stiffness, *k_n_* (N/m)	5 × 10^5^	5 × 10^5^	5 × 10^6^
Tangential stiffness, *k_s_* (N/m)	5 × 10^5^	5 × 10^5^	5 × 10^6^
Internal friction coefficient, *μ*	0.7	0.5	-
Rolling friction coefficient, *μ_r_*	-	1.0	-
Local damping	0.7	0.7	-

**Table 3 materials-14-02909-t003:** Parameters of the normal contact force anisotropy of samples with different RCA contents.

	γ =0%			γ =5%			γ =10%		
	$F_{n}$	$\alpha_{n}$	$\theta_{n}\;({}^{\circ})$	$F_{n}$	$\alpha_{n}$	$\theta_{n}\;({}^{\circ})$	$F_{n}$	$\alpha_{n}$	$\theta_{n}\;({}^{\circ})$
0%	33,210.01	0.68	89.75	41,752.45	0.87	52.79	38,021.11	0.81	61.58
20%	31,801.40	0.65	89.87	40,257.75	0.84	51.91	36,889.23	0.78	58.64
40%	30,605.90	0.64	89.63	39,432.10	0.82	48.12	35,397.66	0.76	57.03
60%	28,967.13	0.64	90.04	37,088.46	0.82	47.48	33,504.11	0.76	56.17
80%	26,722.61	0.62	89.33	36,247.77	0.87	45.25	32,291.18	0.77	55.69

**Table 4 materials-14-02909-t004:** Parameters of the shear contact force anisotropy of samples with different RCA contents.

	γ =0%			γ =5%			γ =10%		
	$F_{t}$	$\alpha_{t}$	$\theta_{t}\;({}^{\circ})$	$F_{t}$	$\alpha_{t}$	$\theta_{t}\;({}^{\circ})$	$F_{t}$	$\alpha_{t}$	$\theta_{t}\;({}^{\circ})$
0%	172.15	0.39	129.14	4867.23	0.92	115.15	3012.45	0.83	118.42
20%	164.06	0.38	130.41	4811.37	0.94	116.48	2890.46	0.82	118.03
40%	134.97	0.46	126.18	4878.95	0.95	117.59	2714.54	0.85	118.91
60%	149.17	0.41	127.07	4857.70	0.96	118.18	2793.37	0.83	119.15
80%	152.78	0.49	130.08	5241.15	0.98	118.86	2675.40	0.84	117.21

## Data Availability

Data is contained within the article.

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
