# Peer review of "Experimental and Numerical Studies on the Direct Shear Behavior of Sand–RCA (Recycled Concrete Aggregates) Mixtures with Different Contents of RCA"

_materials, 2021, doi:10.3390/ma14112909_

Round 1

Reviewer 1 Report

It is an interesting study on the mechanical behavior of RCA-sand mixtures  using the direct shear test and the 3D discrete element
method. Nevertheless, some issues should be clarified:

  • the originality of the research should better described (what is new in the relation to the existing knowledge?) How the results of the study can be used in the engineering practise? Whether the test results are consistent with others findings?
  • the properties of concrete from which RCA are applied should be given
  • what was the humidity of the aggregates?
  • please add the density of sand-RCA mixtures samples
  • the theoretical background of the model applied in discrete element method software PFC3D should be given (constitutive relations, anisotrophy, modeling of contact forces)

Reviewer 3 Report

Review manuscript entitled: “Experimental and numerical studies on the direct shear behavior of sand-RCA (recycled concrete aggregates) mixtures with different contents of RCA”

The manuscript presents experimental and analytical data on direct shear behavior of sand-RCA mixtures with different contents of RCA.

The manuscript should be resubmitted, because it does not follow the format required by the materials journal. You can find the template here :https://www.mdpi.com/files/word-templates/materials-template.dot. Line numbering is necessary.

Furthermore, the text needs proofreading. There are too many grammatical and syntactic errors to mention. Examples: “As a material enormously produced in civil engineering”. “This study presented results”

The authors should mention clearly what are the aims of this manuscript. Do the authors believe that the aims were achieved? Furthermore  they should mention what is the innovative aspect of the study. What is the benefit of the DEM simulation?

Do the authors believe that the findings of this study fill gaps in the state of the art?

Figures 16 and 17 need to be replaced as they have very low quality, and are not easy to read.

I am not sure if the DEM analytical work is actually useful. How can an engineer benefit from this study?

Author Response

Response to Reviewer 3 Comments

Point 1The manuscript presents experimental and analytical data on direct shear behavior of sand-RCA mixtures with different contents of RCA.

The manuscript should be resubmitted, because it does not follow the format required by the materials journal. You can find the template here: https://www.mdpi.com/files/word-templates/materials-template.dot. Line numbering is necessary.

Response 1The authors would like to thank the reviewer for the specific questions raised to improve the quality of the paper. We have revised the manuscript strictly following the format of the materials journal, and the line numbering has also been added.

Point 2: Furthermore, the text needs proofreading. There are too many grammatical and syntactic errors to mention. Examples: “As a material enormously produced in civil engineering”. “This study presented results”

Response 2The authors want to thank the reviewer for raising these questions.

The manuscript has been carefully revised to correct the grammatical and syntactic errors.

Point 3: The authors should mention clearly what are the aims of this manuscript. Do the authors believe that the aims were achieved?  Furthermore, they should mention what is the innovative aspect of the study. What is the benefit of the DEM simulation?

Response 3The authors want to thank the reviewer for pointing out this question.  

(1) We have rewritten the last paragraph of the introduction section to clearly present the aims of the manuscript. We believe that the aims have been achieved.  

(2) The innovative aspect of this study is described as follows:

First, our research is the first to conduct a systematic investigation on the effect of RCA content on the mechanical behavior of sand-RCA mixtures.  The results show that the RCA content plays an important role in the shear behavior of sand-RCA mixtures.

Secondly, it is reported that the shape of the RCA particles is more angular than natural aggregates. In order to take the RCA particle shape into account, we have developed a new method to model authentic RCA particles. The Image-based 3D Object Reconstruction is applied in this method. Comparing to the 3D laser scanning method and CT scanning method, this method is much easier and cheaper.

Finally, the effect of RCA content on the microscopic behavior of sand-RCA mixtures is also comprehensively investigated, and the reason behind the relation between the shear strength and the RCA content is also reported.

(3) The DEM is an excellent tool to explore the microscopic behavior of granular materials. In this study, the effect of RCA content on the contact networks, the distribution of contact orientation, and the distribution of contact forces is presented. And the reason behind the relation between the shear behavior of sand-RCA mixtures and RCA content is also explained based on the analysis of the microscopic results. So we can get a lot of information from a microscopic view, and the microscopic information can help us to understand and explain the macroscopic behavior of granular materials. We think that is the benefit of the DEM simulation.  

Point 4Do the authors believe that the findings of this study fill gaps in the state of the art?

Response 4The authors want to thank the reviewer for raising this question to improve the quality of this manuscript.

Yes, we believe that our findings have filled gaps in the state of the art.

RCA particles are usually used as the coarse grains in the sand-RCA mixtures for landfills. However, the effect of RCA content on the mechanical behavior of the mixtures has not been systematically studied. In this study, we carried out experimental and numerical investigations on this issue. The relation between the RCA content and the shear behavior of sand-RCA mixtures is presented, and the reason is also explained from a microscopic view.

Point 5Figures 16 and 17 need to be replaced as they have inferior quality, and are difficult to read.

Response 5The authors want to thank the reviewer for pointing out this question.

We have changed figures 16 and 17 to improve their quality and make them easier to read.

Point 6: I am not sure if the DEM analytical work is actually useful. How can an engineer benefit from this study?

Response 6 The authors want to thank the reviewer for raising these questions.

The discrete element method (DEM) is an ideal tool to investigate the behavior of granular materials such as gravel and sand. Since Cundall and Strack firstly developed and applied DEM in geotechnical engineering, many researchers devoted lots of effort to improve the method to simulate the behavior of soil and gravel materials. The DEM has advantages in capturing internal force chains and kinematic behaviors of particles.

In this study, we have systematically investigated the effect of RCA content on the shear behavior of sand-RCA mixtures and explored the reason behind the relationship between RCA content and the shear strength of sand-RCA mixtures from a microscopic view. The findings obtained from this study are expected to improve the comprehension of the shear behavior of sand-RCA mixture, which is likely beneficial to the relevant engineering practice.

Round 2

Reviewer 1 Report

I accept the article in the present form

Author Response

The authors want to thank the reviewer for the acceptance of the manuscript. 

Reviewer 3 Report

The manuscript is well written and concise.

I would only suggest the following minor corrections:

Consider replacing Figure 2. - It is not clear.

Figure 14 and 15 I would suggest adding (a) and (b) in the graphs

Author Response

Point 1The manuscript is well written and concise.

Response 1The authors would like to thank the reviewer for the careful review.

Point 2: I would only suggest the following minor corrections:

Consider replacing Figure 2. - It is not clear.

Figure 14 and 15 I would suggest adding (a) and (b) in the graphs

Response 2The authors want to thank the reviewer for raising these questions.

According to the reviewer’s suggestion, the manuscript has been revised as follows

(1) A more clear picture has replaced figure 2.

(2) The (a) and (b) have been added in the graphs of Figures 14 and 15.